# ZIF-8 as a pH-Responsive Nanoplatform for 5-Fluorouracil Delivery in the Chemotherapy of Oral Squamous Cell Carcinoma

**DOI:** 10.3390/ijms25179292

**Published:** 2024-08-27

**Authors:** Jessica Hao, Chider Chen, Kresimir Pavelic, Fusun Ozer

**Affiliations:** 1School of Dental Medicine, University of Pennsylvania, Philadelphia, PA 19019, USA; haoje@upenn.edu; 2Oral and Maxillofacial Surgery, School of Dental Medicine, University of Pennsylvania, Philadelphia, PA 19104, USA; chenc10@upenn.edu; 3Faculty of Medicine, Juraj Dobrila University of Pula, 52100 Pula, Croatia; pavelic@unipu.hr; 4Preventative and Restorative Sciences, School of Dental Medicine, University of Pennsylvania, Philadelphia, PA 19104, USA

**Keywords:** cancer biology, biomaterial(s), drug delivery, cell biology, micro-RNA

## Abstract

5-fluorouracil (5-FU), a chemotherapeutic agent against oral squamous cell carcinoma (OSCC), is limited by poor pharmacokinetics and toxicity. The pH-sensitive zeolite imidazolate framework-8 (ZIF-8) may increase the selectivity and length of 5-FU released into the acidic tumor microenvironment. This study examined the in vitro 5-FU absorption and release profiles of ZIF-8, and then progressed to cytotoxicity assays using the OSCC primary cell line SCC7. The 5-FU loading capacity of ZIF-8 was calculated with UV-vis spectroscopy (λ = 260 nm). 5-FU release was quantified by submerging 5-FU@ZIF-8 in pH 7.4 and 5.5 acetate buffer over 48 h. For the cytotoxicity assays, 5-FU, ZIF-8, and 5-FU@ZIF-8 were added to SCC7 cultures at 25, 50, and 100 μg/mL. Cell viability was assessed through toluidine blue staining and further quantified through transcriptomic RNA sequencing. ZIF-8 stabilized at a maximum absorption of 2.71 ± 0.22 mg 5-FU, and released 0.66 mg more 5-FU at pH 5.5 than 7.4 for at least 72 h. The cytotoxicity assays showed that 5-FU@ZIF-8 had a synergistic inhibitory effect at 50 μg/mL. The RNA sequencing analysis further revealed the molecular targets of 5-FU@ZIF-8 in SCC7. 5-FU@ZIF-8 may release 5-FU based on the pH of the surrounding microenvironments and synergistically inhibit OSCC.

## 1. Introduction

Metastatic and recurrent oral cancer has a dismal prognosis and is commonly caused by well-established risk factors like tobacco use, alcohol consumption, and human papillomavirus (HPV) infection [1,2]. The conventional antimetabolite drug 5-fluorouracil (5-FU) is one of the most effective chemotherapeutic agents against oral squamous cell carcinoma (OSCC) [3,4]. Although 5-FU has therapeutic effects towards a variety of neoplasms [5], it possesses certain caveats that limit its clinical applications in cancer therapy. These include poor pharmacokinetics of the drug, the multidrug resistance of cancer cells, and potential toxicity to healthy tissues [6,7]. Drug delivery systems (DDSs) can serve as a novel strategy to administer anticancer compounds since they can alleviate some of the limitations found in conventional anticancer drugs. DDSs may confer benefits such as improved pharmacokinetics, precise targeting of tumor cells, and the mitigation of side effects [8]. Due to the great potential of DDSs in anticancer drug delivery [8], it is valuable to explore nanoplatforms that may enhance the clinical applications of 5-FU in cancer therapy.

Zeolites are a type of aluminosilicate material regularly distributed micropores ranging between 1 and 10 Å. The pores within their crystalline frameworks can exchange compounds within an aqueous environment, allowing zeolites to serve as carriers of organic compounds and transition metal ions [9]. This unique trait of zeolites is facilitated by other compelling properties such as their high surface area and stability over a broad range of temperatures [10]. Although there are over 40 natural and 230 synthetic zeolites known, their performance is limited by structural stiffness and difficulties in pore size modification [11,12]. To this end, Yaghi et al. proposed and synthesized zeolitic imidazolate frameworks (ZIFs), a type of synthetic MOF composed of Zn(II) or Co(II) copolymerized with imidazolate-type links (Figure 1) [9]. Many such ZIFs may serve as DDSs due to their plethora of positive attributes such as tunable pores and cavities; high aqueous stability; biocompatibility; and, perhaps most importantly, pH-sensitivity [7,12].

What makes ZIFs so enticing as DDSs is their ability to completely degrade and release their contents in an acidic environment while remaining stable under physiological pH [7]. It is well known that through the Warburg Effect [13,14], cancer cells rewire their metabolism to enhance glucose consumption and lactate fermentation [15,16]. This allows cancer cells to grow and proliferate at a rate much higher than that of normal tissues [17,18]. An interesting byproduct of the Warburg Effect is that increased lactate secretion decreases the pH of the tumor microenvironment to 5.5–6.5 [19,20], which is significantly lower than the physiological pH (7.4) of surrounding normal tissues. Targeting these properties through pH-sensitive ZIFs may provide a more targeted way to deliver conventional anticancer therapeutics.

Through a recent systematic review, our team found that there is an increasing number of emerging in vitro and in vivo studies regarding zeolites and ZIFs as effective anticancer drug delivery nanoplatforms. Out of the various types of zeolitic nanoplatforms, ZIF-8 comprised nearly half of the studies relating to cancer therapeutics [7]. Currently, 5-FU has been incorporated into ZIF-8 for enhanced therapeutic effects towards esophageal cancer [21], neuroblastoma [22], and glioblastoma [23]. Due to the positive results from these studies [8], we believe it would be valuable to extend the application of 5-FU@ZIF-8 to oral cancer therapeutics. In the present study, the in vitro uptake and release of 5-FU in ZIF-8 was quantified, and the ensuing results were applied to SCC7 (murine oral squamous cell carcinoma) assays.

## 2. Results

### 2.1. Loading and Release of 5-FU@ZIF-8

In the present study, commercially available ZIF-8 was shown to be capable of absorbing and releasing 5-FU in a controlled manner. A standard concentration curve of 5-FU in dH_2_O with 10% methanol, with a line of best fit of absorbance of 44.27 [concentration of 5-FU] + 2.88, was created (Figure 2A,B). With this serving as a comparison, a 5-FU loading curve was constructed over a period of 56 h. At 48 h, the ZIF-8 reached a maximum absorption of 2.71 ± 0.22 mg of 5-FU and remained stable at 56 h with 2.75 ± 0.19 mg (Figure 2C). Upon being exposed to 1 M acetate buffer solution, 5-FU@ZIF-8 showed a drastic difference in its drug release capabilities in the pH 7.4 and 5.5 solutions. The carriers at pH 7.4 released an average of 0.66 mg more 5-FU than at pH 5.5, and maintained their well-demarcated difference for at least 72 h (Figure 2D).

### 2.2. In Vitro Cell Compatibility against SCC7 Cell Line

A cytotoxicity study was conducted in triplicate with several concentrations of 5-FU, ZIF-8, and 5-FU@ZIF-8 (Figure 3). Compared to the control, 5-FU had a slight inhibitory effect at all concentrations. ZIF-8 had an inhibitory effect at 100 μg/mL, but did not impact the cells at lower concentrations of 50 μg/mL and 25 μg/mL. 5-FU@ZIF-8 treatments had the most significant inhibitory effect, most evident at 50 and 100 μg/mL. This suggests a potential synergistic effect of the two compounds. Based on the cell viability staining results, we chose 50 μg/mL as the optimum concentration for further experimentation because this would minimize the confounding impact of ZIF-8 while maximizing the impact of 5-FU@ZIF-8.

### 2.3. RNA Sequencing Analysis Revealed Molecular Targets of the DDS

Finally, RNA sequencing analysis was performed to further reveal the molecular targets of the various test groups in SCC7. We found that 841 transcripts were significantly overexpressed after 5-FU treatment and 763 transcripts were increased after 5-FU@ZIF-8 treatment (Figure 4A,B). A correlation analysis showed both 5-FU alone and 5-FU@ZIF-8 treatment significantly altered RNA expression levels in SCC7 (Figure 4C). An enrichment analysis of gene oncology (GO) showed that the 5-FU treatment group had significantly decreased cell cycle genes, DNA repair capability, mitochondrial respiration capability, and fatty acid metabolism. In addition, the group had an upregulated cell killing process, as well as upregulated protein deacetylase activity and H3K4 methylation, indicating that the 5-FU treatment decreased protein and DNA activity in SCC7 cells (Figure 4D). Cell proliferation genes and cellular metabolic pathways are largely decreased after 5-FU@ZIF-8 treatment. 5-FU@ZIF-8 treatment upregulated co-factor effects and increased cell senescence and apoptosis (Figure 5).

## 3. Discussion

### 3.1. The Trial and Error behind the Loading and Release of 5-FU@ZIF-8

The authors began by studying how the type of carrier, heat activation, type of solvent, ratio of 5-FU to ZIF-8, immersion time, and release medium impacted the loading and release experiments (Table 1). A key breakthrough within the present study was the necessity of heat-activating the ZIF-8 prior to usage. This improved the outcomes drastically by allowing the ZIF-8 to absorb up to 271.41 ± 21.94 mg/g of 5-FU. Through the obtained data, it was observed that ZIF-8 had similar 5-FU loading capacities in both dH_2_O and PBS, while having little to no loading capacity within methanol. In addition, the optimum absorption occurred when 5-FU and ZIF-8 at a 1:1 ratio was immersed for more than 48 h. These findings were used to construct a unique new protocol for the absorption and release of the drug from the carrier.

In this study, ZIF-8 was chosen as the representative nanoplatform due to its widespread use in anticancer research and availability on a commercial level [7]. However, through a Gibbs ensemble Monte Carlo (GEMC) simulation, Proenza et al. found that 5-FU was not evenly distributed within ZIF-8. Although the innermost pores of ZIF-8 were easily penetrated by solvents like methanol and water, they were inaccessible to larger drugs such as 5-FU. The absorbed drug molecules remained within the outermost pores of the ZIF-8 since diffusion between neighboring pores was significantly inhibited by high-energy barriers [24,25].

In comparison to ZIF-8, ZIF-90 has higher sorption kinetics and desorption enthalpy due to its smaller particle size, which results in greater surface area, mesoporosity, and an increased number of polar groups. ZIF-90 is composed of hydrophilic 2-carboxaldehyde imidazolate linkers instead of the hydrophobic 2-methylimidazole linkers that make up ZIF-8. In addition to ZIF-8, the authors intended to use ZIF-90 as a representative DDS. However, without some of the necessary chemical synthesis equipment in our lab, it was difficult to produce a ZIF-90 product free from contaminants. With some further searching, ZIF-90 was also found to be commercially available online from Chemsoon Co (Shanghai, China). A future direction of this project is to replicate the experiment with the commercial ZIF-90, and compare the results to those of ZIF-8.

In addition to the synthetic ZIFs, clinoptilolite, which is a naturally derived zeolite, was tested for drug loading and release. Clinoptilolite is an aluminosilicate mineral that has garnered much attention in biomedicine due to its anti-inflammatory, detoxifying, and antioxidant properties [26]. However, clinoptilolite was found to be unable to absorb 5-FU, even after heat activation. Therefore, after a literature search and learning about the differences between zeolites and ZIFs, the authors determined that ZIFs were better suited for this study.

Perhaps the most interesting comparison was the dissolution of ZIF-8 in various solvents. When designing the original proposed methodology, the authors referenced 5-FU release studies in which PBS served as the release medium [27]. However, after numerous trials, it was discovered that the commercially available ZIF-8 was unable to degrade in PBS, even when the pH was as low as 3.3. This is demonstrated by the remaining presence of crystals in the pH 7.4 and pH 3.3 PBS solution after 1 month. A possible explanation for this finding is that while commercially available ZIF-8 was utilized for the present study, the referenced articles synthesized their own zeolitic carriers with unique modifications. Discrepancies in the synthesis procedures between the studies may have contributed to variations within the dissolution characteristics. Upon returning to the literature, it was found that a pioneering article by Sun et al., who first synthesized pure ZIF-8 as a 5-FU DDS without additional modifications, utilized a pH 5 acetate buffer to simulate the acidic environment [12]. Enlightened by the methods from their article, the authors of the present study immersed ZIF-8 in pH 4, 5.5, 6.5, and 7.4 acetate buffer solutions. While crystals were still present in the pH 6.5 and 7.4 acetate buffer solutions at 1 month, the ZIF-8 dissolved within minutes in the pH 4 and 5.5 acetate buffer solutions.

The previous comparison revealed unique qualities between ZIF-8 and acetate that may bolster the effectiveness of ZIF-8 as an anticancer DDS. As the precursor to acetyl-CoA, acetate ions are ubiquitous within the biological environment and are commonly used for the synthesis of fatty acids and cholesterol molecules in the cell membrane [28,29]. Rapidly dividing cancer cells have an avidity for cholesterol and lipids, which can be fulfilled through increased exogenous lipid consumption or elevated endogenous lipogenesis and cholesterol synthesis [30,31]. Similar to other cancers, OSCC may reprogram metabolism by increasing de novo lipid synthesis and β-oxidation [32,33,34]. For example, fatty acid synthase (FAS), which is responsible for the formation of long-chain fatty acids from malonyl-CoA and acetyl-CoA, is overexpressed in OSCC [35]. Increased FAS gene expression is foreshadowed by hypoxia and acidification of the tumor microenvironment [36], the latter of which is the original intended target of ZIF-8. Therefore, the authors believe that ZIF-8 may serve as a promising carrier for drug delivery to OSCC microenvironments due to its dissolution in both an acidic pH and an acetate-buffered environment.

### 3.2. In Vitro Cell Culture Studies and RNA Sequencing Analysis

In the cytotoxicity assays, there was a potential synergistic effect between 5-FU and ZIF-8 that was most evident at 50 μg/mL. At this concentration, the confounding impact of ZIF-8 towards the cells was minimized, while the combined impact of 5-FU@ZIF-8 was maximized. These findings were supported by RNA sequencing, which showed that 5-FU@ZIF-8 treatment decreased cell proliferation genes and metabolic pathways and increased cell senescence and apoptosis. From the present study’s results, we hypothesize that long-term treatment with 5-FU@ZIF-8 nanoparticles can inhibit cancer stem cells and achieve a chronic therapeutic effect.

Further cellular experiments will observe the effects of 5-FU@ZIF-8 towards mesenchymal stem cells (MSCs). It is well known that cancer does not merely reside in transformed epithelium, but also metastasizes to cells relating to the immune system and mesenchyme [37,38]. MSCs are overabundant in OSCC, facilitating tumor growth by impacting stromal development and hindering the systemic immune response [39]. Therefore, observing the differences in the effects of 5-FU@ZIF-8 towards MSC and SCC7 cell lines will be crucial in understanding the holistic impact of 5-FU@ZIF-8 towards OSCC. Promising results from these future studies will lead to the transplantation of these cells into mice for in vivo tumor inhibition studies.

## 4. Materials and Methods

### 4.1. Materials

5-Fluorouracil (5-FU, ≥99%) and zeolitic imidazolate framework-8 (≥99%) were purchased from Sigma Aldrich (St. Louis, MO, USA). The ZIF-8 was heat-activated for 24 h at 160 °C using a laboratory oven (Binder ED056) from LabRepCo (Horsham, PA, USA) before use. UV-vis absorption spectroscopy was performed on a Thermo Scientific NanoDrop 2000 photospectrometer (Philadelphia, PA, USA).

DMEM/RPMI-1640, 200 mM glutamate, penicillin/streptomycin, and TrypLE Express were purchased from Thermo Fisher Scientific (Philadelphia, PA, USA). Fetal Bovine Serum was obtained from Hyclone Laboratories (Logan, UT, USA).

### 4.2. Creation of a 5-FU Standard Curve

A 5-FU stock solution was prepared and diluted to yield various concentrations, and the UV–visible absorbance of each solution was analyzed at λ = 260 nm. The results were plotted against their respective concentrations to yield a basis of comparison for future UV-vis experiments.

### 4.3. Synthesis of 5-FU-Loaded ZIF-8

A solution containing 10 mg 5-FU in 5 mL of dH_2_O with 10% methanol was prepared. The initial absorbance of the solution was determined by UV-vis spectroscopy at λ = 260 nm. After adding 10 mg activated ZIF-8, the resulting mixture was placed on an orbital shaker at 25 °C for 48 h to allow the components to combine. UV-vis measurements were taken periodically throughout the absorption phase to estimate the amount of 5-FU absorbed into the porous nanocarrier. The drug loading capacity was calculated through the following equation:*Loaded Drug Capacity* = [*Initial Concentration* − *Remaining Concentration*] × *Solvent Volume*(1)

Finally, the solution was centrifuged at 12,000× rpm for 8 min. The supernatant was discarded, and the pellet was dried at 50 °C for 12 h.

### 4.4. In Vitro Quantification of 5-FU Release

The in vitro release of 5-FU from loaded ZIF-8 was assessed by UV-vis spectroscopy at λ = 260 nm. The loaded particles were submerged into 5 mL of pH 7.4 and pH 5.5 acetate buffer and maintained at 37 °C in a shaking incubator. Over 48 h, the concentrations of 5-FU within the acetate buffer were determined by UV-vis spectroscopy at λ = 260 nm.

### 4.5. Cell Culturing

SCC7 (murine oral squamous cell carcinoma) was cultured in DMEM/RPMI-1640 supplemented with 10% (*v*/*v*) FBS, 1% (*v*/*v*) 200 mM glutamate, and 1% (*v*/*v*) penicillin/streptomycin. Immediately after extraction from liquid nitrogen storage, the frozen SCC7 culture tube was placed in a 37 °C water bath for 1 min. The thawed cells were pipetted into 9 mL of premade culture medium and centrifuged at 1300× rpm for 5 min. After centrifugation, the supernatant was replaced with 1 mL of fresh medium to resuspend the pellet. The cells were transferred to a cell culturing dish, and an additional 9 mL of medium was added. The dish was incubated at 37 °C until the cells reached 100% confluence at 7 days. On days 3 and 6, the medium was gently suctioned out and 10 mL fresh medium was added.

On day 7, the cells were passed onto two additional dishes for further proliferation. Upon removing the medium, the plate was washed with PBS. A total of 3 mL TrypLE Express was added to the washed cells, which were then incubated for 5 min at 37 °C and briefly agitated until the cells were visibly detached. A total of 3 mL fresh medium was added to the plate to protect the cells from further trypsin digestion. The culture was then centrifuged at 1300× rpm for 5 min, and the pellet was resuspended in 1 mL fresh medium. The number of cells per plate was estimated using a hemocytometer to be around 19,275,000 cells. A small amount of resuspended cells was added to each plate containing 10 mL of fresh medium. The dishes were incubated at 37 °C in an atmosphere of 5% CO_2_ until the cells reached 100% confluence.

### 4.6. In Vitro Cell Compatibility Studies

The in vitro cytotoxicity of 5-FU, ZIF-8, and 5-FU@ZIF-8 was assayed against SCC7 cells and visualized via a nuclear staining technique. Each treatment was conducted in triplicate. The prepared SCC7 cells were passed to a 12-well tissue culture plate at a density of 8.0 × 10^5^ cells per well and incubated overnight at 37 °C and 5% CO_2_. 5-FU, ZIF-8, and 5-FU@ZIF-8, which were autoclaved for 90 min at 120 °C to ensure sterility, were added to the culture medium at concentrations of 25 μg/mL, 50 μg/mL, and 100 μg/mL. The drug-incorporated media were then added to the wells to replace the original culture media. After 72 h of incubation, the drug-incorporated media were removed, and 1 mL toluidine blue stain was added to each well. The plate was placed on an orbital shaker for 8 h, after which the dye was suctioned out and the wells were washed with PBS and air-dried for 24 h.

### 4.7. RNA Sequencing Methods

The prepared SCC7 cells were passed to a 6-well tissue culture plate and incubated overnight at 37 °C and 5% CO_2_. 5-FU and 5-FU@ZIF-8 were added to the culture medium at 50 μg/mL. For each group, this process was performed in triplicate. After 72 h, the cells were detached with trypsin and centrifuged. The supernatant was discarded, and the remaining cell pellet was frozen at −20 °C. Each sample group was prepared for RNA extraction using the QIAzol Lysis Reagent protocol, and subsequent sample analysis was conducted by Novogene.

## 5. Conclusions

In the present study, commercially available ZIF-8 was shown to be capable of adsorbing 5-FU in a controlled manner up to 271.41 ± 21.94 mg/g. The release of 5-FU from the carrier was medium and pH-dependent, and occurred best in a pH 5.5 acetic acid buffer solution. Finally, 5-FU@ZIF-8 may inhibit murine OSCC proliferation in a synergistic manner, with the optimum concentration being 50 μg/mL. Recent studies have demonstrated the increased avidity of OSCC towards endogenous lipogenesis, which utilizes acetate as a precursor in the synthesis process. The increased de novo fatty acid synthesis and acidification of the tumor microenvironment may represent key targets for ZIF-8 drug delivery.

## Figures and Tables

**Figure 1 ijms-25-09292-f001:**
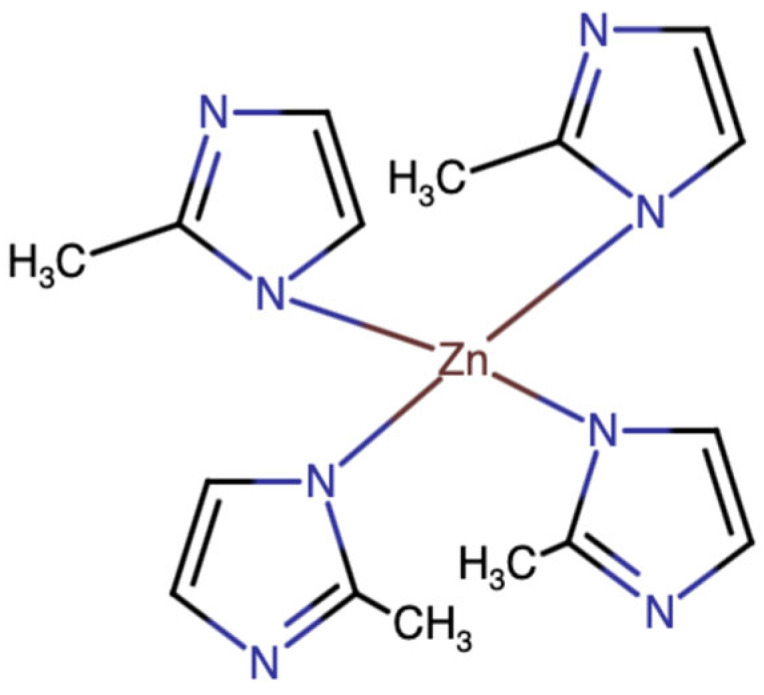
Chemical structure of zeolitic imidazolate framework-8 (ZIF-8).

**Figure 2 ijms-25-09292-f002:**
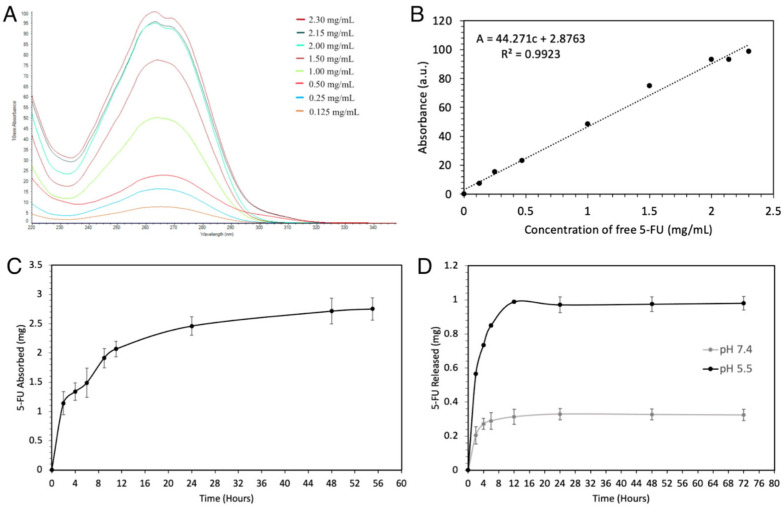
Loading and releasing curves of 5-FU in ZIF-8. (**A**) The 260 nm UV-vis spectra of 5-FU in dH_2_O with 10% methanol. (**B**) Standard curve of 5-FU in dH_2_O with 10% methanol, as determined by UV-vis spectra. (**C**) Loading of 5-FU in ZIF-8 in dH_2_O with 10% methanol. (**D**) A comparison of 5-FU delivery in various acidities of 1 M acetate buffer solution.

**Figure 3 ijms-25-09292-f003:**
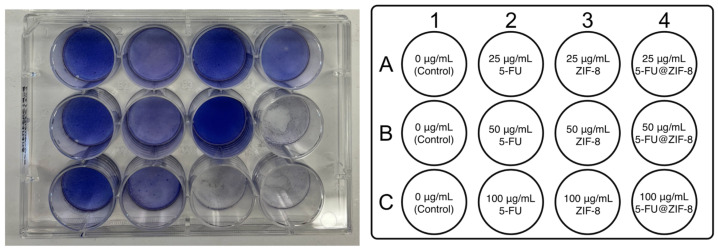
Various concentrations of 5-FU, ZIF-8, and 5-FU@ZIF-8 were added for SCC7 cell compatibility studies. Each treatment was conducted in triplicate.

**Figure 4 ijms-25-09292-f004:**
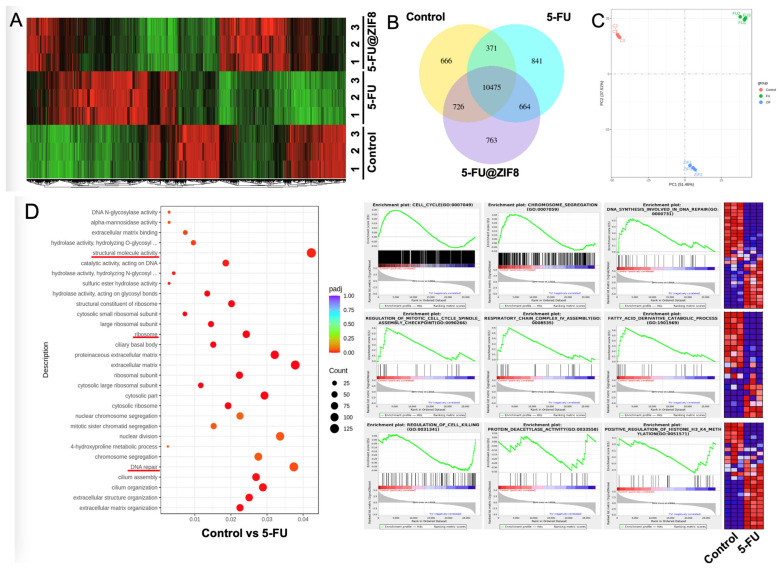
RNA sequencing analysis comparing the treatment groups. (**A**,**B**) Overexpression of certain transcripts was seen. (**C**) Correlation analysis. (**D**) Enrichment analysis of gene oncology. Abbreviated content include “hydrolase activity, hydrolyzing O-glycosyl compounds” and “hydrolase activity, hydrolyzing N-glycosyl compounds”.

**Figure 5 ijms-25-09292-f005:**
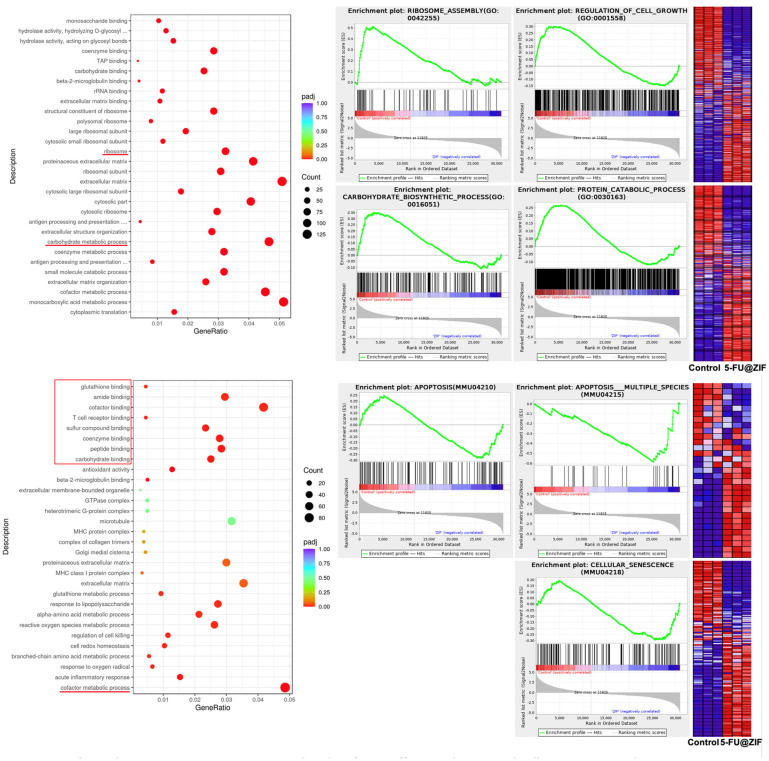
RNA sequencing analysis comparing the control and 5-FU@ZIF-8 treatment groups. Abbreviated content include “hydrolase activity, hydrolyzing O-glycosyl compounds”, “antigen processing and presentation of peptide antigen via MHC class I”, and “antigen processing and presentation of endogenous antigen”.

**Table 1 ijms-25-09292-t001:** The absorption of 5-FU was optimized by varying absorption parameters such as the type of heat-activated carrier, presence of heat activation, 5-FU/ZIF-8 ratio, and solvent.

Variables of Absorption		Loaded Drug Capacity (mg/g)
Type of heat-activated carrier	Clinoptilolite	No change
	ZIF-8	271.41 ± 21.94
	ZIF-90	No change
Heat activation of ZIF-8	160 °C for 24 h	271.41 ± 21.94
	None	No change
5-FU/ZIF-8 ratio	1/3	101.42 ± 10.24
	1/2	150.99 ± 2.11
	1/1	271.41 ± 19.07
	4/1	248.38 ± 9.84
Solvent	dH_2_O	200.00 ± 1.95
	PBS	179.68 ± 21.73
	Methanol	No change

## Data Availability

The raw data supporting the conclusions of this article will be made available by the authors on request.

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
