# Peer review of "ZIF-8 as a pH-Responsive Nanoplatform for 5-Fluorouracil Delivery in the Chemotherapy of Oral Squamous Cell Carcinoma"

_ijms, 2024, doi:10.3390/ijms25179292_

Round 1
Reviewer 1 Report
Comments and Suggestions for Authors
This is a good paper, describing a practical support for the use of chemotherapeutical agent in cancer therapy.
Some minor corrections might contribute to the quality of the paper.
row 47 "zeolites are a type of metal-organic frameworks" - Zeolites are not metal-organic polymers, but inorganic (aluminosilicates). During research on syntheic aspects of zeolites some variants with organic molecules, fragments have been produced, which can be regarded as "metal-organic" polymers (frameworks).
row 49 the Authors here mention a "review" without reference. This should be added.
rows 287/288 "absorbing" would be better as "adsorbing"
row 299 "within" a strange expression - "in" or "using" would be better
The commercial product ZIF-8 has a central role in the research described in the paper. A short description of its chemical nature and/or a sketch of its structure could help to the non-specialist reader.
Comments on the Quality of English LanguageSome remarks are described in the previous box.
Author Response
Thank you for your constructive comments towards our paper. We appreciate you taking the time to review.
- row 47 "zeolites are a type of metal-organic frameworks" - Zeolites are not metal-organic polymers, but inorganic (aluminosilicates). During research on syntheic aspects of zeolites some variants with organic molecules, fragments have been produced, which can be regarded as "metal-organic" polymers (frameworks).
- Thank you for your clarification on the topic. The wording has been changed to “aluminosilicate material”
- row 49 the Authors here mention a "review" without reference. This should be added.
- Row 49 does not mention a review. Did the reviewer intend to point out a different line?
- rows 287/288 "absorbing" would be better as "adsorbing"
- The correction has been made.
- row 299 "within" a strange expression - "in" or "using" would be better
- Row 299 does not contain “within”; however, the wording in row 289 has been changed to “in”
- The commercial product ZIF-8 has a central role in the research described in the paper. A short description of its chemical nature and/or a sketch of its structure could help to the non-specialist reader.
- Thank you for your advice. A diagram of ZIF-8 has been added to row 58 and relabeled as fig 1.
Reviewer 2 Report
Comments and Suggestions for Authors
This manuscript is missing a lot of important points that must be addressed before it can be considered for publication:
1. Figure 1, for the loading capacity of 5-FU, mg/g is recommended as the unit. While for the release of 5-FU, mg/g of release percent (%) is better to provide.
2. Figure 2, it is better to provide a quantitative result. Moreover, has the test been repeated for each treatment?
3. Section 3.1, Line 150-154, do the authors try ZIF-90 as another platform for 5-FU?
Line 158-160, the results of clinoptilolite to load 5-FU need to provide.
4. “5-FU has been incorporated into ZIF-8 for enhanced therapeutic effects” (Line 73), it is suggested that a comparison with representative literature be aadded in the discussion section
5. Section 4.1, “The SEM images were captured through a ThermoFisher FEI Quanta 250 FEG Scanning Electron Microscope.” (Line 220). Where are the SEM results?
6. Line 218, “The ZIF-8 was heat activated °C using …”. What is the temperature?
7. Ref. 39 is incomplete, while Ref. 22 was not found in the manuscript.
Comments on the Quality of English Language
good
Author Response
Thank you for your constructive comments towards our paper. We appreciate you taking the time to review.
- Figure 1, for the loading capacity of 5-FU, mg/g is recommended as the unit. While for the release of 5-FU, mg/g of release percent (%) is better to provide.
- Thank you for your suggestion. After discussion, we decided to keep the units the same to easily compare the graphs.
- Figure 2, it is better to provide a quantitative result. Moreover, has the test been repeated for each treatment?
- Each treatment was conducted in triplicate. This specification was added to lines 111 and 271.
- Section 3.1, Line 150-154, do the authors try ZIF-90 as another platform for 5-FU?
- The authors hope to try ZIF-90 as another platform for 5-FU as a future direction of research, but this manuscript will solely focus on ZIF-8.
- Line 158-160, the results of clinoptilolite to load 5-FU need to provide.
- The amount of 5-FU loaded by clinoptilolite is 0 mg/g.
- “5-FU has been incorporated into ZIF-8 for enhanced therapeutic effects” (Line 73), it is suggested that a comparison with representative literature be added in the discussion section
- Thank you for your meaningful suggestion. After review, it is found to be difficult to compare our paper with the pre-existing 5-FU and ZIF-8 literature due to the diversity of delivery methods used. For example, Ref 22 uses protein-titanocene complexes while Ref 23 uses hyaluronic acid-drug conjugates.
- Section 4.1, “The SEM images were captured through a ThermoFisher FEI Quanta 250 FEG Scanning Electron Microscope.” (Line 220). Where are the SEM results?
- SEM was attempted at our school’s cell & molecular biology core, but 5-FU was too small to be seen through SEM. An image of ZIF-8 was captured, but the authors deemed that this does not contribute much to the manuscript. Therefore, this sentence was deleted from the manuscript upon review.
- Line 218, “The ZIF-8 was heat activated °C using …”. What is the temperature?
- Thank you for catching this typo that was omitted while copying the manuscript. The ZIF-8 was heat activated for 24 h at 160℃. This has been added to the manuscript.
- 39 is incomplete, while Ref. 22 was not found in the manuscript.
- The in-text citation of ref. 22 has been updated. This is the literature regarding ZIF-8’s effects towards neuroblastoma.
- 39 has been updated.
Reviewer 3 Report
Comments and Suggestions for Authors
In this manuscript, the authors prepared a ZIF-8 based drug delivery system as a pH-responsive nanoplatform for 5-fluorouracil delivery in the chemotherapy of oral squamous cell carcinoma.After a careful review, I think this manuscript should be rejected. I have identified several significant shortcomings that need to be addressed before the manuscript can be considered for publication.
Firstly, the details of the material preparation are insufficient, and the characterizations are not clear. A more comprehensive description of the synthesis process and detailed characterization data are essential for readers to understand and reproduce the work.
Secondly, the biological experiments are severely lacking. There is an insufficiency in cell trials and the absence of animal experiments. Comprehensive biological evaluations, including in vivo studies, are crucial to validate the biological implications and potential applications of the materials.
Finally, the interpretation of the RNA sequencing results is inadequate. A more in-depth and comprehensive analysis is needed to draw meaningful conclusions and provide clear insights into the underlying mechanisms.
Author Response
Thank you for your constructive comments towards our paper. We appreciate you taking the time to review.
- Firstly, the details of the material preparation are insufficient, and the characterizations are not clear. A more comprehensive description of the synthesis process and detailed characterization data are essential for readers to understand and reproduce the work.
A more detailed characterization data can be found in section 2.1. Reading section 4.3 along with 2.1 may help the reader better understand the material preparation process. Since the ZIF-8 was commercially available, the synthesis process simply involved adding ZIF-8 and 5-FU together in solution for it to combine on its own.
2. Secondly, the biological experiments are severely lacking. There is an insufficiency in cell trials and the absence of animal experiments. Comprehensive biological evaluations, including in vivo studies, are crucial to validate the biological implications and potential applications of the materials.
This paper consists of cell experiments that will serve as the basis for further cell trials and animal experiments. It serves as preliminary research that brings the ZIF/5-FU system to the field of oral cancer research. Our lab hopes to advance to more comprehensive biological evaluations and in vivo studies in the coming year.
3. Finally, the interpretation of the RNA sequencing results is inadequate. A more in-depth and comprehensive analysis is needed to draw meaningful conclusions and provide clear insights into the underlying mechanisms.
A more in-depth and comprehensive analysis has been provided in line 114. Figure 4 has been updated to reflect the changes.
Round 2
Reviewer 2 Report
Comments and Suggestions for Authors
None
Reviewer 3 Report
Comments and Suggestions for Authors
This manuscript can be accepted. No other comments.